# Genetic evidence for a fall-spawning group of Gulf sturgeon (*Acipenser oxyrinchus desotoi*) in the Apalachicola River, Florida, USA

Jacob O. Zona[1¤*], Brian R. Kreiser[1], Adam J. Kaeser[2], Adam G. Fox[3], Mark J. D'Ercole[3]

1 School of Biological, Environmental, and Earth Sciences, University of Southern Mississippi, Hattiesburg, Mississippi, United States of America, 2 Panama City Fish and Wildlife Conservation Office, United States of America Fish and Wildlife Service, Panama City, Florida, United States of America, 3 Warnell School of Forestry and Natural Resources, University of Georgia, Athens, Georgia, United States of America

¤ Current address: Department of Natural Resource Management, South Dakota State University, Brookings, South Dakota, United States of America
* jozona@svsu.edu

## Abstract

The Gulf sturgeon (*Acipenser oxyrinchus desotoi*) is a large, long-lived, anadromous fish inhabiting the northern Gulf of Mexico. This charismatic fish was hunted to near extinction in the early 1900s. In 1991 the subspecies was listed as threatened under the Endangered Species Act. Recovery continues to be challenged by threats such as habitat destruction, fisheries bycatch, and climate change. There are seven known natal rivers. Historically, each river was thought to contain a single, spring-spawning group. Recent studies have discovered several rivers (Suwannee, Choctawhatchee) contain a second, fall-spawning group. This study utilizes genetic techniques to investigate the proposed existence of a fall-spawning group in the Apalachicola River, Florida. Juvenile Gulf sturgeon were sampled between May and July, from 2013 to 2022. Four adults were also captured on the spawning grounds during October of 2022. Samples were genotyped for thirteen microsatellite loci to assess genetic population structure within the Apalachicola River. Analyses detected two distinct genetic groups ($F_{ST} = 0.085$). Dates of capture, length frequency distributions of juveniles, and genetic assignment of spawning adults allowed us to characterize these as separate spring- and fall-spawning groups. While approximately 90% of juveniles collected were assigned to the spring, only slight differences in genetic diversity were detected between groups. The temperature window for spawning was found to be three weeks shorter on average in the fall than the spring, highlighting the need for additional research into differing environmental or anthropogenic influences on these populations. The discovery of a fall-spawning group of Gulf sturgeon in the Apalachicola River improves our understanding of the representation, redundancy, and resiliency of the species and provides critical information for improved management of this river system.

**Data availability statement:** All relevant data are within the manuscript and its Supporting Information files.

**Funding:** This study was made possible by a partnership between the U.S. Fish and Wildlife Service, the University of Southern Mississippi, and the University of Georgia. Funding for this work was provided by the U.S. Fish and Wildlife Service, U.S. Army Corps of Engineers, University of Georgia Warnell School of Forestry and Natural Resources, and by the Open Ocean Restoration Area Trustee Implementation Group of the Deepwater Horizon Trustee Council as part of their Final Restoration Plan 1 for birds and sturgeon. The findings and conclusions in this article are those of the authors and do not necessarily represent the views of the U.S. Fish and Wildlife Service. The funders had no role in study design, data collection and analysis, decision to publish, or preparation of the manuscript.

**Competing interests:** The authors have declared that no competing interests exist.

## Introduction

The Gulf sturgeon (*Acipenser oxyrinchus desotoi*: Acipenseriformes, Acipenseridae) is a large, anadromous fish native to the northern Gulf of Mexico [1]. Subadults and adults overwinter in the nearshore waters of the gulf, while juveniles spend the winter in the lower salinity estuaries. Each spring all age groups make the migration back into freshwater systems where they spend the remainder of the year [2–4]. Gulf sturgeon reach sexual maturity late in life (8–12 years) and females are suspected to have spawning intervals of multiple years [5]. This suite of life history traits makes them particularly sensitive to anthropogenic disturbances, and the subspecies was listed as 'Threatened' under the Endangered Species Act in 1991 [6]. Historic over-fishing as well as habitat alteration and indirect fisheries mortality have led to range contraction, local extirpations, and severe declines in abundance [5,7–9].

Reproducing populations of Gulf sturgeon are now restricted to seven river systems. Historically, each was thought to be represented by a single group that spawned during the spring [6]. In 2012, Randall and Sulak [10] suggested the Suwannee River supported a fall-spawning group based on evidence from adult sturgeon movements and sexual characteristics, as well as the size structure of the juvenile population. The capture of several unexpectedly small juveniles in 2013 and again in 2019 was subsequently interpreted by Dula et al. [11] as potential evidence of fall-spawning in the Apalachicola River. Spawning runs of Atlantic sturgeon (*Acipenser oxyrinchus oxyrinchus*), a closely related congener, have been documented in both spring and fall seasons in several southern-latitude, Atlantic Coast systems [12–15]. Molecular analyses have found these intra-drainage stocks to be genetically distinct [16–19]. A more comprehensive understanding of the genetic structure and timing of spawning within each Gulf sturgeon population is critical for effective management of the species, especially in regulated river systems where access to spawning habitat and hydrologic conditions during spawning might influence overall reproductive success [20–22].

The Apalachicola River is the largest river in Florida by discharge [23]. Its watershed, referred to as the Apalachicola-Chattahoochee-Flint River Basin (ACF), drains approximately 5 million hectares (ha) of the Gulf Coastal Plain and Appalachian Mountains [24]. The ACF system is fragmented by numerous dams, five of which are major projects operated by the U.S. government (i.e., Army Corps of Engineers). Jim Woodruff Lock and Dam, completed in 1957 at the confluence of the Flint and Chattahoochee rivers, created Lake Seminole reservoir and permanently blocked access to approximately 78% of historic Gulf sturgeon habitat in the ACF system (Fig 1) [6]. Only 50–80 ha of potentially suitable spawning habitat exists in the Apalachicola below the dam [25,26]. Concern for the potential impact of hydrologic operations on Gulf sturgeon spawning motivated studies in the 2000s that involved egg collections and modeling of the effects of recruitment failure on the trajectory of the Apalachicola population [27–29]. Results of these studies suggested reduction in total area of spawning habitat during low flow conditions, and/or short-term variations in discharge resulting from hydropeaking operations, may periodically result in reduced or failed recruitment. Through consultation, this work led to changes in the way the Army

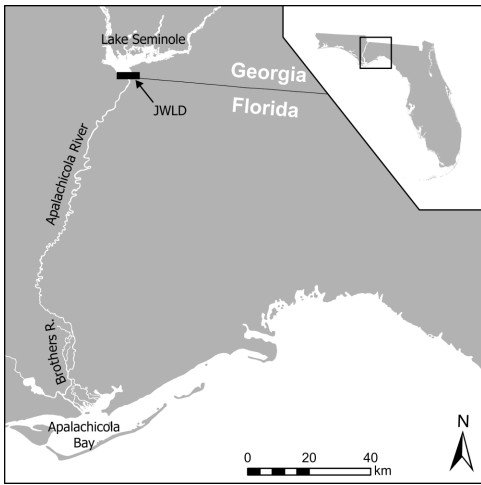

**Fig 1. Map of the Apalachicola River, Florida, USA.** Gulf sturgeon were sampled for this study from the Apalachicola River below the Jim Woodruff Lock and Dam (JWLD) and the Brothers River tributary.

Corps of Engineers managed flows during the spring [30]. Fall-spawning in this system was not yet documented at the time of these consultations and thus was not taken into consideration.

The existence of a fall-spawning group of Gulf sturgeon in the Apalachicola River would have important management implications for this population, where recruitment appears to be relatively low [31]. Environmental variables such as temperature and flow are anthropogenically influenced and may differ seasonally [32], warranting a closer look at their relationship with spawning season. Knowledge of fall-spawning would also contribute insights to studies of juvenile age, growth, and recruitment, and help inform the overall conservation status of the species. Thus, the purpose of this study was to use genetic techniques to investigate the purported existence of a fall-spawning group in the Apalachicola River and to assess its level of genetic distinctiveness from the spring group.

## Materials and methods

### Field sampling

Gulf sturgeon were sampled in the Apalachicola River from May through July of 2013–2022 by field teams from the University of Georgia and the U.S. Fish and Wildlife Service. All sampling was conducted following protocols set by the Institutional Animal Care and Use Committee Permits (A2019 01–002-Y3-A3, A2021 09–010-Y3-A3) and annual collections permits issued by the Florida Fish and Wildlife Conservation Commission to the U.S. Fish and Wildlife Service (e.g., 2024 FNW-005, FNW23−05, FNW22−03, FNW21−09, FNW20−06, FNW19−08). Fish were captured using anchored monofilament gill nets (45.7 x 3 meters) comprised of three equal-length panels of 7.6-, 8.9-, and 10.2-centimeter mesh stretch measure. Identical nets were used in previous studies to effectively capture juvenile Gulf sturgeon in the Apalachicola River [11,31]. Nets were soaked for 60–120 minutes, depending on conditions. Each weekday during the sampling season, 6–12 nets were set. Sampling was concentrated in the Brothers River, a coastal plain tributary of the Apalachicola River where Gulf sturgeon commonly aggregate; sampling also occurred in various reaches of the mainstem Apalachicola River (Fig 1) [31]. All captured sturgeon were scanned for tags and, if not previously tagged, implanted with a unique passive integrated transponder tag. Each fish was measured, and a small (approximately 1 cm²) tissue sample was taken from its anal fin for genetic analysis. Fish were then released back into the river near their site of capture. Tissue samples were stored in 95% ethanol at room temperature and shipped to the University of Southern Mississippi for analysis.

## Laboratory procedures

All laboratory procedures were conducted in agreement with Institutional Animal Care and Use Committee Protocol #17101202. Total genomic DNA was extracted from tissue samples using the DNeasy Tissue Kit (Qiagen, Inc., Valencia, California). Prior to amplification, DNA quality and concentration were verified for each sample using a NanoDrop Spectrophotometer (Thermo Fisher Scientific Inc., Waltham, Massachusetts). Thirteen microsatellite loci were amplified (Atlantic sturgeon – *Aox*B34, *Aox*D32, *Aox*D44, *Aox*D54, *Aox*D64, *Aox*D165, *Aox*D170, *Aox*D188, *Aox*D234, *Aox*D241, *Aox*D242, and *Aox*D297 [33] and lake sturgeon - *LS*68 [34]) using 12.5 µL polymerase chain reactions (PCR) (S1 Table). These loci are functionally diploid and have been used in previous molecular studies [33–35,47,48]. PCR components consisted of 1x *Taq* PCR buffer (New England Biolabs, Ipswich, Massachusetts), 2.5 mM $MgCl_2$, 200 µM dNTPs, 0.25 units of *Taq* polymerase, 0.16 µM of M13 tailed forward primer [36], 0.16 µM of M13 tailed reverse primer, 0.08 µM of M13 labeled primer (Eurofins, Inc., Louisville, Kentucky), 10–200 ng/µL of template DNA, and nuclease-free water to the final volume. Polymerase chain reaction thermocycling conditions were as follows: Initial denaturing step of 94 °C for 2 min followed by 35 cycles of 30 sec at 94 °C, 1 min at 53–58 °C (S1 Table), and 1 min at 72 °C, followed by a final elongation step of 10 min at 72 °C. Genotypes were visualized using a LI-COR 4300 DNA sequencer (LI-COR Inc., Lincoln, Nebraska).

## Data analysis

Research suggests movement between river systems is limited in juvenile Gulf sturgeon, particularly age-1 and younger [2,9]. Fin ray aging work with juveniles in the Apalachicola River indicated age-1 fish ranged from 390 to 520 millimeters (mm) fork length (FL) [37]. Only fish ≤520 mm FL were included in the analyses to ensure our dataset contained only fish spawned within this river system.

The program STRUCTURE v. 2.3.4 [38] is widely used to identify discrete genetic groupings and was used to characterize population genetic structure of Gulf sturgeon within the Apalachicola River. STRUCTURE uses a Bayesian approach to assign individuals, based on their multi-locus genotypes, to some number of different genetic groups (K) that are in Hardy-Weinberg and linkage equilibrium. An admixture model was run under the assumption of correlated allele frequencies between groups with population of origin information as a prior [39]. Values of K between 1 and 6 were tested, each with 20 iterations of 150,000 Markov chain Monte Carlo repetitions with burn-ins of 100,000. The most likely value of K was then determined by examining the average likelihood scores for each value of K and the Δ K analysis as performed by StructureSelector [40]. CLUMPP 1.1.2 [41] was used to summarize the STRUCTURE results across all iterations for the most likely value of K. Distruct 1.1 [42] was used to create a bar chart of admixture proportions. Length frequency distributions for each genetic group were graphed in R v. 4.4.0 to help visualize differences in spawning season and birth year [43].

As part of a different study, during October of 2022, four adult Gulf sturgeon were captured in spawning condition in the Apalachicola River. Milt or eggs were identified during handling and/or acoustic transmitter implantation. These individuals were genotyped based on the methods above and were added to the dataset. An additional STRUCTURE analysis was run under the same parameters to determine if these adults assigned to a distinct genetic group, presumably representing a fall spawn.

Pairwise $F_{ST}$ between genetic groups was determined using the "hierfstat" package in R to provide insight into the degree of genetic divergence [44]. Calculation of 95% confidence intervals was conducted using 1,000 bootstrap samples. Several measures of genetic diversity were calculated for each group. GenAlEx v. 6.5 [45,46] was used to determine the average number of alleles per locus ($N_a$), observed heterozygosity ($H_o$), and expected heterozygosity ($H_e$). Allelic richness ($A_R$) was also calculated using "hierfstat." Differences between groups were tested for statistical significance using two-tailed, paired sample t-tests using loci as paired samples. If assumptions necessary for parametric tests were not met, Wilcoxon signed-rank tests were used. Goodness-of-fit to the normal distribution was assessed using the Shapiro-Wilk test and equal variance was assessed using the Bartlett's test. All statistical tests were conducted in R v. 4.4.0 [43].

To evaluate when fall spawning might be occurring in the Apalachicola River and compare spring vs. fall spawning windows, we examined temperature records from the USGS Chattahoochee gage 02358000 located approximately 1 km upstream of the primary spawning site [29]. Complete temperature records were available from 2017 to 2023. These records were used to determine the total number of days in each spawning season with temperatures suitable for spawning, between 17.0 and 25.0 °C [7].

## Results

### Dataset description

A total of 526 juvenile Gulf sturgeon were sampled from the Apalachicola River system during spring and summer (April 23 through July 29) of the years 2013–2022. Forty-five fish were excluded from the study, leaving a sample size of 481 individuals for analysis (S2 Table). Of the excluded samples, four had missing data at more than three microsatellite loci, three belonged to individuals sampled twice in different years, and thirty-eight were from individuals larger than our 520 mm FL threshold.

### STRUCTURE analysis

The STRUCTURE analysis suggested there were two distinct genetic groups within the Apalachicola River (Fig 2), based on the results of the ΔK analysis (S1 Fig). The average probability of ancestry (mean q-score)- or probability of belonging to a genetic group- was high for both groups (0.98 and 0.93; Table 1), indicating gene flow between groups is limited. However, several samples from each group possessed q-scores that indicated mixed ancestry, including seven fish with q-scores of less than 0.70 for their primary group of origin. Each of the remaining 474 juvenile Gulf sturgeon were assigned to one genetic group or the other.

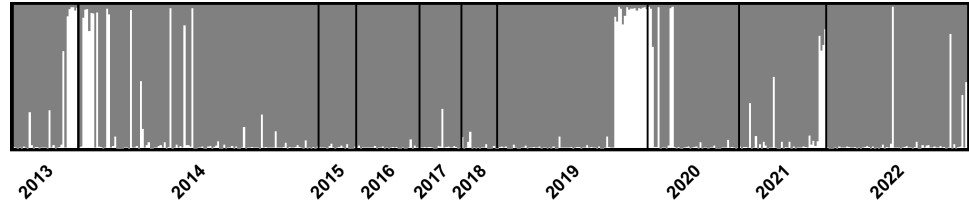

**Fig 2. Results of STRUCTURE analysis at K = 2 for Apalachicola River Gulf sturgeon.** Vertical bars represent individual fish where the proportion of each color represents the degree of ancestry to the corresponding genetic group, with gray for the spring-spawning group and white for the fall-spawning group. Samples are grouped by capture year and sorted by fork length from largest (left) to smallest (right). All fish were ≤520 mm FL and collected from 2013–2022.

**Table 1. Genetic comparisons of the spring- and fall-spawning groups of Gulf sturgeon in the Apalachicola River.**

|  | Spring | Fall |
|---|---|---|
| q-score | 0.983 (0.003) | 0.926 (0.012) |
| $N_a$ | *9.8 (1.6) | *6.8 (1.0) |
| $A_R$ | 7.8 (1.2) | 6.8 (1.0) |
| $H_o$ | 0.66 (0.05) | 0.65 (0.06) |
| $H_e$ | 0.66 (0.05) | 0.68 (0.05) |

Average probability of ancestry (q-score) at K = 2, number of alleles ($N_a$), allelic richness ($A_R$), and observed and expected heterozygosity ($H_o$ and $H_e$) are reported with standard error in parentheses. * denotes statistically significant difference (p = 0.007).

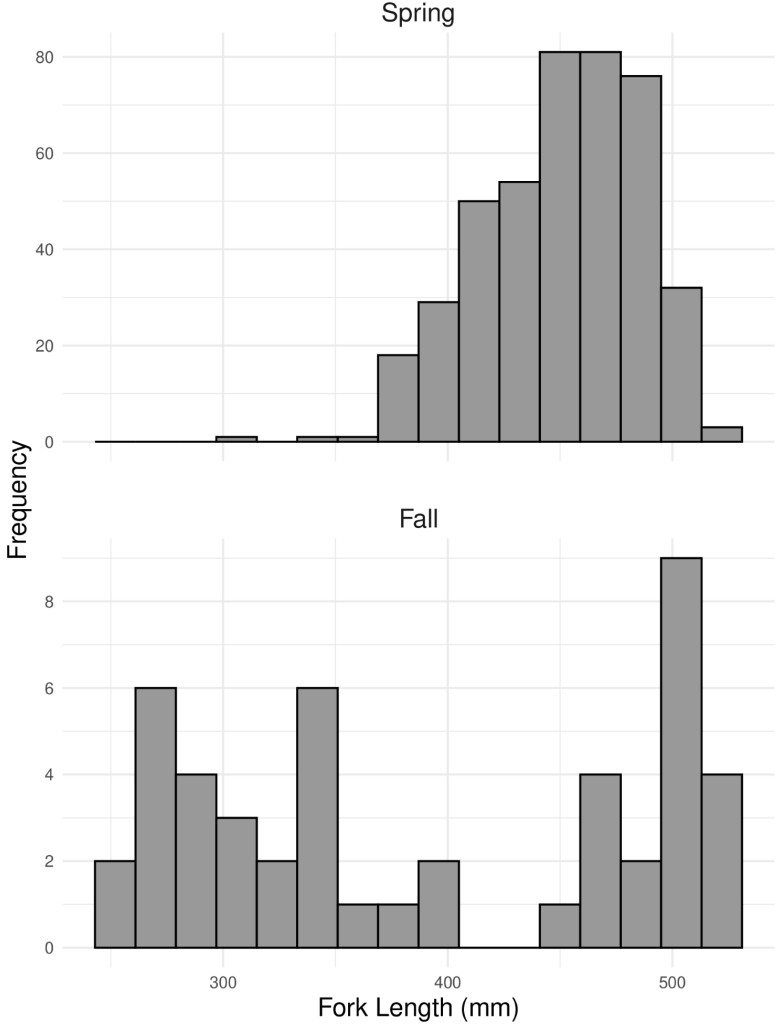

## Designation of genetic groups by spawning season

Fourteen Gulf sturgeon ≤300 mm FL were captured across multiple sampling years. Through STRUCTURE analysis, thirteen of these individuals were assigned to Group 2 and one was suggested to be of mixed ancestry. This size class was previously inferred to have originated from a fall spawning event [11]. Additionally, all four of the adults captured in spawning condition during the fall were assigned to Group 2 during the second STRUCTURE analysis (mean q-score = 0.97). Using these fish as references, we classified individuals from Group 2 as fall spawned and those from Group 1 as spring-spawned. The two groups of juveniles exhibited distinctly different length frequency distributions; fish classified as spring-spawned exhibited a unimodal distribution, whereas fall-spawned fish demonstrated a bimodal distribution (Fig 3). During the summer capture period, the average fish in the spring-spawned group was 451 mm FL and 95% of individuals were at least 390 mm FL (Fig 3). In that same period, approximately half of the fall-spawned fish were 250–400 mm FL (Group 2A), and the other half were >450 mm FL (Group 2B). Based on the size, dates of collection, and timing of favorable

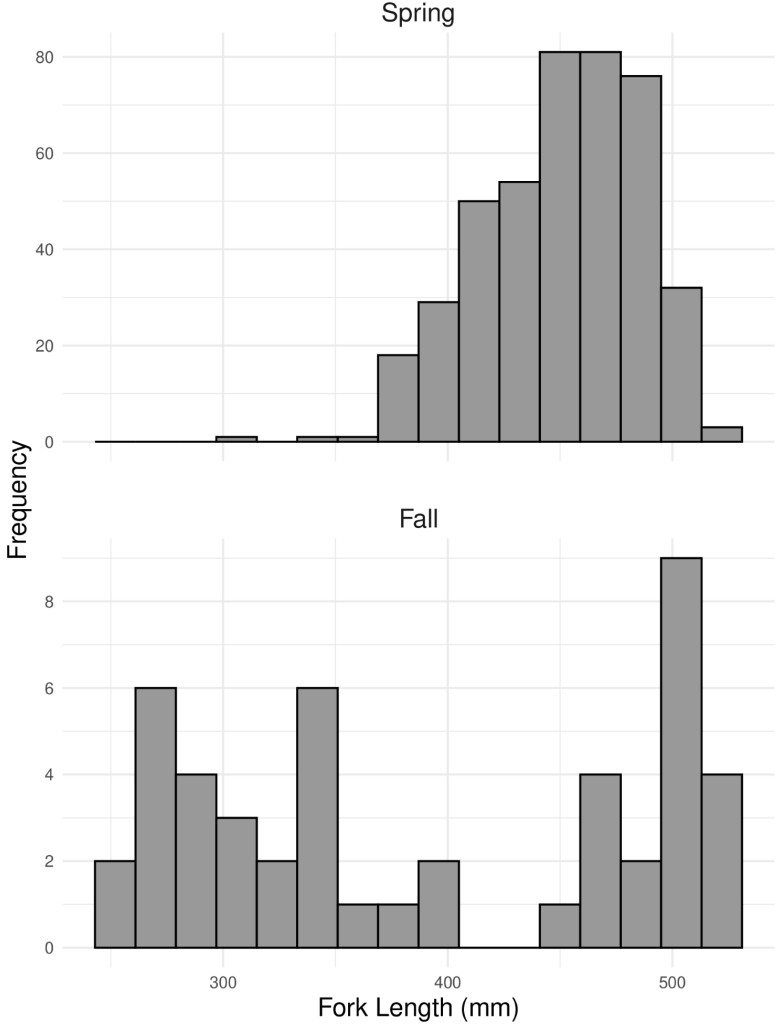

**Fig 3. Juvenile length frequency distributions for spring- and fall-spawning groups of Gulf sturgeon in the Apalachicola River.** All fish included are ≤ 520 millimeters in fork length and were sampled from 2013–2022. Bimodal distribution of the fall group represents two distinct age classes, age-0 and age-1.

temperatures for spawning in the Apalachicola system [25], fish belonging to Group 2A were classified as spawned during the fall prior to their collection (i.e., actual age range 7–10 months old), and fish belonging to Group 2B were classified as spawned two fall seasons prior to collection (i.e., age range 19–22 months old). The fish of Group 1 were classified as spawned during spring of the year prior to their collection (i.e., age range 12–14 months).

## Comparison of spring- and fall-spawning groups

Assignment of each sturgeon in the spring and fall groups to a birth year revealed distinct differences in the likelihood of encountering spring- vs. fall-spawned individuals through our sampling efforts (Table 2). Across all study years, 90% of fish (n = 427) were classified as spring-spawned. We observed a cohort of spring-spawned fish in every year of the study. In contrast, only 47 fall-spawned individuals (10%) were collected; these fish were observed in only four out of ten years (Table 2). Most fall-spawned fish identified in this study (89%) were determined to have originated from spawning events in 2012 (n = 19) and 2018 (n = 23) (Table 2). Seven fish of mixed ancestry were not assigned to a birth year and season during this analysis.

Calculation of pairwise $F_{ST}$ found substantial genetic differentiation between the spring and fall groups ($F_{ST}$ = 0.085, 95% CI = 0.050–0.119). The number of alleles per locus ($N_a$) and allelic richness ($A_R$) were higher on average in the spring, however, only $N_a$ was found to be statistically significant ($N_a$: p = 0.007, V = 64; $A_R$: p = 0.094, V = 70). Measures of heterozygosity were similar between groups ($H_o$: p = 0.9, t = 0.14, df = 12; $H_e$: p = 0.6, V = 38) (Table 1).

Between 2017 and 2023, temperatures favorable for spawning during the spring occurred 64 days on average (range 52–75) between the months of March and May. Favorable temperatures for fall spawning occurred 43 days on average (range 25–64) between the months of October and November.

## Discussion

This paper provides the most substantial evidence to date for the existence of a fall-spawning group of Gulf sturgeon in the Apalachicola River. Several lines of evidence from this and previous studies, including molecular analyses, length-frequency distributions of juveniles, and adults identified on spawning grounds or captured in spawning condition, indicate that fall spawning exists within the three easternmost rivers supporting Gulf sturgeon: the Apalachicola,

**Table 2. Birth year and spawning season for juvenile Apalachicola River Gulf sturgeon.**

| Birth Year | Spring n | Fall n | Total | Spring % | Fall % |
|---|---|---|---|---|---|
| 2012 | 26 | 19 | 45 | 58 | 42 |
| 2013 | 107 | 0 | 107 | 100 | 0 |
| 2014 | 19 | 0 | 19 | 100 | 0 |
| 2015 | 32 | 0 | 32 | 100 | 0 |
| 2016 | 21 | 0 | 21 | 100 | 0 |
| 2017 | 18 | 0 | 18 | 100 | 0 |
| 2018 | 59 | 23 | 82 | 72 | 28 |
| 2019 | 40 | 0 | 40 | 100 | 0 |
| 2020 | 38 | 4 | 42 | 90 | 10 |
| 2021 | 67 | 1 | 68 | 99 | 1 |
| Total | 427 | 47 | 474 | 90 | 10 |

All fish were sampled from 2013–2022 and ≤520 mm FL. Samples were sorted into birth year and spawning season cohorts (Spring n and Fall n) by genetic group assignment and length frequency analysis. The proportion of samples represented by each season is also reported (Spring % and Fall %). Seven fish of mixed ancestry were not classified and are not included in this table.

the Choctawhatchee [47, S. Rider unpublished data], and the Suwannee Rivers [10, Price et al. In Press]. Considering the prevalence of dual spawning among southern Atlantic sturgeon populations [15,18,19], it is possible that additional fall-spawning groups of Gulf sturgeon in other rivers remain to be identified.

The two spawning groups of Gulf sturgeon in the Apalachicola River were found to be genetically distinct, with an $F_{ST}$ comparable to the values reported when comparing populations from adjacent rivers [35,48]. Spring- and fall-spawning groups within the Choctawhatchee and Suwannee rivers exhibited similar levels of genetic differentiation to those reported in this study [47,48]. These findings mirror those for the Atlantic sturgeon, where the amount of genetic differentiation between groups that spawned in different seasons was comparable to that found between groups that occupied different rivers [18]. Gulf sturgeon are known to have high river-of-origin fidelity [3,8,9,35], which has led to considerable reproductive isolation between river systems [49,50]. The amount of genetic differentiation we observed between spring- and fall-spawning groups in this study suggests Gulf sturgeon also demonstrate high spawning season fidelity.

Fall-spawned juveniles were detected less frequently and in much lower overall numbers than spring-spawned juveniles. Despite this, only slight differences in genetic diversity were found between groups. The higher number of alleles found per locus in the spring could be attributed to a larger sample size as allelic richness was not significantly different between groups. Because age-1 juveniles were prioritized for this study, some discrepancy in the number of spring- vs. fall-spawned fish in the sample set may be explained by lower net vulnerability, as mesh sizes may have been too large for effective capture of fish in Group 2A (i.e., fall fish <400 mm FL). However, fish in Group 2B (i.e., fall fish >450 mm FL) would likely have exhibited net vulnerability similar to the spring-spawned juveniles. When considering only fish in this size range (450–520 mm FL), the proportion of fall-spawned juveniles is even lower (10% vs 8%). In the Suwannee River, spring-spawning Gulf sturgeon are found in larger numbers than fall-spawners, although the discrepancy is narrower, with the two groups accounting for approximately 65% and 35%, respectively [Price et al In Press].

Environmental factors that influence Gulf sturgeon recruitment success and juvenile abundance may fluctuate seasonally or via human influence [7,29,30]. For example, this study found that the temporal window for favorable spawning temperatures in the Apalachicola River is approximately 3 weeks shorter on average during fall. However, the primary focus of this study was to verify the purported existence of a fall-spawning group in this system. These comparisons serve to highlight the need for further research into the population trends of different spawning groups and their potential drivers.

The presence of a fall-spawning population of Gulf sturgeon should be considered in the context of research into juvenile dynamics in the Apalachicola River, conservation status assessments, and future management decisions including operation of the Jim Woodruff Lock and Dam. Adjustments to water release aimed at improving spawning success, similar to those made for the spring-spawning population [25,30], may be necessary in fall as well. The results of this study point to the merits of pairing genetics analyses with traditional aging techniques to more accurately assign juvenile fish to an exact birth season and year of origin. This knowledge should benefit studies that aim to model and compare the growth of juvenile sturgeon cohorts, both within and across populations. The ability to parse spring- from fall-spawned individuals also provides an opportunity to investigate patterns of recruitment success among spawning groups over time. Although additional work is necessary to determine whether environmental factors influence or favor the success of one spawning group over another, the ability to accurately classify juveniles to a birth year and season will remain a cornerstone of these investigations.

With regard to conservation status, the discovery of temporal isolation within the Apalachicola River elevates the representation, or breadth of genetic diversity, known to exist within this species [51]. This fall-spawning group may contain an important set of unique genes and traits that aid in adaptation to a changing environment in the future [52]. Beyond genetic diversity, the existence of the fall-spawning group also contributes crucial conservation elements of redundancy and resiliency [51], as first noted by Dula et al. [11]. These authors hypothesized that fall-spawning adults may have escaped a major mortality event caused by Hurricane Michael by remaining upriver in reaches that did not suffer hypoxia. Our study confirms that this fall 2018 spawn contributed to the number of juveniles captured in subsequent years. Given

the importance of redundancy, resiliency, and representation to the overall conservation status and recovery of Gulf sturgeon, future investigations should focus on identifying additional fall-spawning groups, evaluating factors supporting their existence, and enacting conservation measures to protect these stocks across the full range of the species.

## Supporting information

**S1 Table. Microsatellite loci used to genotype Gulf sturgeon in this study.** Loci developed by May et al. 1997 and Henderson-Arzapolo and King 2002. Repeat motif, forward and reverse primer sequence, PCR annealing temperature, and GenBank accession number listed for each locus.
(TXT)

**S2 Table. Juvenile Apalachicola River Gulf sturgeon dataset.** Table includes data used for analysis in this study, including microsatellite genotypes, capture date, and fork length (mm) for each sample.
(TXT)

**S1 Fig. ΔK analysis suggesting K = 2 for Apalachicola River Gulf sturgeon.** All samples were ≤520 mm FL and collected from 2013–2022. To identify the most likely number of genetic groups, ΔK analysis determines the rate of change in log likelihood of the STRUCTURE data between each value of K.
(TIF)

## Acknowledgments

Thank you to Robbilyn Verges for assistance with genetic data collection, Jake Schaefer (University of Southern Mississippi) for graduate mentorship, and numerous technicians (University of Georgia) for their contributions in the field. This work is made possible through the continued collaboration of many partners at the U.S. Fish and Wildlife Service, National Marine Fisheries Service, Florida Fish and Wildlife Conservation Commission, University of Georgia, University of Southern Mississippi, Louisiana State University, and University of Florida.

## Author contributions

**Conceptualization:** Jacob O. Zona, Brian R. Kreiser, Adam J. Kaeser.

**Data curation:** Jacob O. Zona.

**Formal analysis:** Jacob O. Zona, Brian R. Kreiser.

**Funding acquisition:** Brian R. Kreiser, Adam J. Kaeser, Adam G. Fox.

**Investigation:** Jacob O. Zona, Adam J. Kaeser, Adam G. Fox, Mark J. D'Ercole.

**Methodology:** Jacob O. Zona, Brian R. Kreiser, Adam J. Kaeser, Adam G. Fox, Mark J. D'Ercole.

**Project administration:** Brian R. Kreiser, Adam J. Kaeser, Adam G. Fox.

**Resources:** Brian R. Kreiser, Adam J. Kaeser, Adam G. Fox, Mark J. D'Ercole.

**Software:** Jacob O. Zona, Brian R. Kreiser.

**Supervision:** Jacob O. Zona, Brian R. Kreiser, Adam J. Kaeser.

**Validation:** Jacob O. Zona, Brian R. Kreiser.

**Visualization:** Jacob O. Zona, Brian R. Kreiser, Adam J. Kaeser.

**Writing – original draft:** Jacob O. Zona, Brian R. Kreiser, Adam J. Kaeser, Adam G. Fox, Mark J. D'Ercole.

**Writing – review & editing:** Jacob O. Zona, Brian R. Kreiser, Adam J. Kaeser, Adam G. Fox, Mark J. D'Ercole.

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
