## [Decision Letter · Decision Letter 0]

17 Mar 2025

PONE-D-25-04780Genetic evidence for a fall spawning group of Gulf sturgeon (*Acipenser oxyrinchus desotoi* ) in the Apalachicola River, FloridaPLOS ONE

Dear Dr. Zona,

Thank you for submitting your manuscript to PLOS ONE. After careful consideration, we feel that it has merit but does not fully meet PLOS ONE’s publication criteria as it currently stands. Therefore, we invite you to submit a revised version of the manuscript that addresses the points raised during the review process.

We look forward to receiving your revised manuscript.

Kind regards,

Sayyed Mohammad Hadi Alavi

Academic Editor

PLOS ONE

Journal Requirements:

2. Thank you for stating the following financial disclosure: [This study was made possible by a partnership between the U.S. Fish and Wildlife Service, the University of Southern Mississippi, and the University of Georgia. Funding for this work was provided by the U.S. Fish and Wildlife Service, U.S. Army Corps of Engineers, University of Georgia Warnell School of Forestry and Natural Resources, and by the Open Ocean Restoration Area Trustee Implementation Group of the Deepwater Horizon Trustee Council as part of their Final Restoration Plan 1 for birds and sturgeon. The findings and conclusions in this article are those of the authors and do not necessarily represent the views of the U.S. Fish and Wildlife Service.]. 

Additional Editor Comments:

Dear Dr Jacob O. Zona: Thank you very much for your submission to PLOS ONE. We have completed evaluation of your MS with helps of three reviewers. Please revise your MS considering their comments and those of mine appended below, and submit the revised MS.

Very best regards

SMH Alavi, AE

Please include a map for sampling location. It needs to follow PLOS ONE criteria when you use a map in your publication.

Please include details of microsatellite loci: accession numbers and, primer sequences, and optimum concentrations.

L47: Please add family and order

L51: You may correct as “similar to most sturgeon species”. There are species with lower age of maturity.

L97: Is it possible to indicate approximate water temperature during the period of spawning?

Please cite supplementary materials as Figure S1 or Figure S2.

Please add name of species in the title of table and/or legend of figures.

Reviewers' comments:

Reviewer's Responses to Questions

**Comments to the Author**

1. Is the manuscript technically sound, and do the data support the conclusions?

Reviewer #1: Yes

Reviewer #2: Partly

Reviewer #3: Yes

2. Has the statistical analysis been performed appropriately and rigorously? 

Reviewer #1: Yes

Reviewer #2: Yes

Reviewer #3: Yes

3. Have the authors made all data underlying the findings in their manuscript fully available?

Reviewer #1: Yes

Reviewer #2: Yes

Reviewer #3: Yes

4. Is the manuscript presented in an intelligible fashion and written in standard English?

Reviewer #1: Yes

Reviewer #2: No

Reviewer #3: Yes

5. Review Comments to the Author

Reviewer #1: The paper presents important findings that contribute to our understanding of Gulf sturgeon population structure and spawning behavior. With the suggested revisions, the manuscript will be strengthened and more accessible to a broader audience. I recommend minor revisions before acceptance

Reviewer #2: PONE-D-25-04780

The authors provide a description of Genetic evidence for a fall spawning group of Gulf sturgeon (*Acipenser oxyrinchus desotoi* ) in the Apalachicola River, Florida. The MS presents English grammatical problems. The authors should correct the language to improve readability. The MS is written with separate ideas that are not articulated together. The abstract sentences are long and should be shortened for better understanding. The introduction presents problems in establishing the background of the topic. It has not segmented the different parts of the Materials and Methods section, which needs to be corrected. It should also specify the sampling section, primer information, and DNA quality. The results aren’t discussed continuously. The discussion is not focused on describing the problem, however, there is a weak and disorganized discussion of the topic.

The MS needs to be rewritten to order the ideas clearly.

I have provided some clarifying comments to help improve the MS, I will explain that the comments are not language corrections because this requires expert work.

Abstract:

Line 22: In this section, "1900’s"the comma should be removed.

Line 26: "have been shown to contain" can be changed to "were shown to contain"

The abstract sentences are long and should be shortened for better understanding.

The abstract section does not properly explain the sampling and testing steps, and is ambiguous.

Introduction:

Line 63: "was" should be changed to "were".

Line 74 "river basin" should be capitalized

Line 79: "and" should be changed to "as well as"

Line 90: basis should be changed to "as the basis"

Line 93: "the" recovery

Line 110: by "the" international

The introduction presents problems in establishing the background of the topic.

Methods:

Line 132: "in accordance with" should be changed to "in agreement with"

Line 144:'initial' should be capitalized

Line 151: individuals should have a comma after it

Line 152: "the" analysis

Line 154: "the" population

Line165: "was" should be changed to "were"

Line 168: in "a" spawning condition

Line 169: a coma should be added before during

Line 177: "the" average number

It has not segmented the different parts of the Materials and Methods section, which needs to be corrected. It should also specify the sampling section, primer information, and DNA quality.

Results:

Line 209: "a" degree

Line 209: "the" corresponding

Line 210: "within" should be removed before "by"

Line 228: "a" should be changed to "the"

Line 233: "the" should be added before "size"

Line 261: "were not" should be changed to "weren't"

Line 228: spring-spawned

Line 229: fall-spawned

The results aren’t discussed continuously.

Discussion:

Line 293: fall-spawned

Line 293: spring-spawned

Line 296: there shouldn't be a "the" before fall

Line 305: fall-spawned

Line 310: fall-spawned

Line 314: fall-spawned

Line 316: "a" larger sample

Line 330: fall-spawned

Line 345: "to" should be changed to "of"

The discussion is not focused on describing the problem, however, there is a weak and disorganized discussion of the topic.

Reviewer #3: I liked this paper and the approach taken. That said, I am not an expert on genetics--so this question may be naive. Does ploidy have an impact on assessment of rigour of your group assignments? Do you view Gulf sturgeon as 2N or 4N? What is the impact of a putative chromosome count of 120 on your interpretation of your FST values? I think a mention of this would help. Does a higher ploidy/chromosome count strengthen your argument?

Line 168: you referred to a study from 2022 that 4 adult sturgeon were captured in spawning condition. Were they running milt and/or expressing eggs? Clarification of this would be helpful.

6. PLOS authors have the option to publish the peer review history of their article (what does this mean? ). If published, this will include your full peer review and any attached files.

**Do you want your identity to be public for this peer review?** For information about this choice, including consent withdrawal, please see our Privacy Policy .

Reviewer #1: **Yes: ** Dr. Shahid Mahmood

Reviewer #2: No

Reviewer #3: No

---

## [Author Response · Author response to Decision Letter 1]

2 May 2025

We greatly appreciate your efforts to help us improve the content and clarity of this manuscript. Below are our responses to editor and reviewer comments.

“Add "Florida, USA" for geographic specificity”

- Revised

“Briefly mention the genetic techniques used (e.g., microsatellite analysis) and the specific implications for conservation management.”

- Revised. Added a description of the genetic techniques to the abstract. Management implications are described in the final sentence.

“Briefly mention the genetic tools used in similar studies on other sturgeon populations (e.g., Atlantic sturgeon) to set the stage for the current study.”

- Revised. Mentioned that similar genetic tools and analyses are used in previous studies within the methods and discussion.

“Add a sentence or two about the potential genetic implications of the Jim Woodruff Lock and Dam on population structure.”

- The JWLD does not isolate populations above and below the dam, it prevents populations from existing above. Gulf sturgeon are anadromous and cannot complete their life cycle above the dam, so there is no population above. Mentioned in lines 74-77.

“Include a brief justification for the specific mesh sizes used in the gill nets, as this could influence the size range of captured fish.”

- Lines 111-114 provide justifications for mesh sizes.

“Add a brief explanation of why the specific microsatellite loci were chosen and their relevance to Gulf sturgeon.”

- Revised (Lines 134-135)

“Include a brief explanation of why STRUCTURE was chosen over other population structure analysis tools.”

- Revised (Lines 152-153)

“Provide a brief explanation of why these exclusions were necessary (e.g., missing data, size thresholds).”

- Lines 194-197 provide this information

“Add a brief discussion of the potential implications of the mixed ancestry individuals (q-scores < 0.70).”

- Revised (Lines 203-204)

“Add a brief comparison with other rivers (e.g., Suwannee, Choctawhatchee) to provide broader context.”

- Revised. Information added to discussion paragraphs 1 and 2.

“Include a brief mention of potential environmental factors (e.g., temperature, flow) that may influence recruitment in the fall.”

- Revised (Lines 312-318)

“Add a brief mention of the potential long-term implications of lower genetic diversity in the fall-spawned group.”

- No differences in genetic diversity were determined to be both statistically significant and biologically relevant. Only number of alleles, which could be a function of larger sample size, was significantly higher in the spring group. Lines 299-302.

“Include a brief discussion of how the findings could inform future dam operations or habitat restoration efforts.”

- Revised (Lines 320-325)

“Add a few more recent studies on Gulf sturgeon genetics or fall spawning in other species for broader context.”

- Revised. Added a citation for a recent Atlantic sturgeon population genetics paper that came out in the last several months.

“Add a brief caption explaining the significance of the gray and white bars.”

- Explained in lines 210-212

“Include a brief caption explaining the significance of the bimodal distribution in the fall-spawned group.”

- Revised (Lines 245-246)

“Add a brief footnote explaining the statistical significance of the differences between groups.”

- Revised (Line 220)

“Provide a brief explanation of how the data in S1 Table were used in the analysis.”

- Revised (Lines 538-540)

“Please include a map for sampling location. It needs to follow PLOS ONE criteria when you use a map in your publication.”

- Revised, See Figure 1

“Please include details of microsatellite loci: accession numbers and, primer sequences, and optimum concentrations.”

- Revised, See Supplemental Table 1

“L47: Please add family and order”

- Revised

“L51: You may correct as “similar to most sturgeon species”. There are species with lower age of maturity.”

- Revised

“L97: Is it possible to indicate approximate water temperature during the period of spawning?”

- Revised

“Please cite supplementary materials as Figure S1 or Figure S2.”

- Revised

“Please add name of species in the title of table and/or legend of figures.”

- Each table and figure referencing Gulf sturgeon contains the name in the title.

“Line 22: In this section, "1900’s"the comma should be removed.”

- Revised

“Line 26: "have been shown to contain" can be changed to "were shown to contain"”

- Revised

“The abstract sentences are long and should be shortened for better understanding.

The abstract section does not properly explain the sampling and testing steps, and is ambiguous.”

- Revised. Sentences were shortened for better understanding and the content was edited to better explain sampling and analysis.

“Line 63: "was" should be changed to "were".”

- The verb is agreeing with “capture” not “juveniles”

“Line 74 "river basin" should be capitalized”

- Revised

“Line 79: "and" should be changed to "as well as"

- No real grammatical error. Authors believe “and” reads easier.

“Line 90: basis should be changed to "as the basis"”

- Revised, removed during organizational review

“Line 93: "the" recovery”

- Revised, removed during organizational review

“Line 110: by "the" international”

- Revised

“The introduction presents problems in establishing the background of the topic.”

- Revised. Edits to the content and organization of the introduction were made to better establish project background and objectives.

“Line 132: "in accordance with" should be changed to "in agreement with"”

- Revised

“Line 144:'initial' should be capitalized”

- Revised

“Line 151: individuals should have a comma after it”

- Revised

“Line 152: "the" analysis”

- Revised

“Line 154: "the" population”

- Revised

“Line165: "was" should be changed to "were"”

- Revised

“Line 168: in "a" spawning condition”

- The term spawning condition indicates one option of a binary, either the fish is in the condition that indicates spawning or it is not.

“Line 169: a coma should be added before during”

- Revised

“Line 177: "the" average number”

- Revised

“It has not segmented the different parts of the Materials and Methods section, which needs to be corrected. It should also specify the sampling section, primer information, and DNA quality.”

- Revised. Different parts of the materials and methods were segmented. Sampling locations were described in the field sampling section, a supplemental table (Table S1) of primer information was added, and methodology for verification of DNA quality was added.

“Line 209: "a" degree”

- Revised

“Line 209: "the" corresponding”

- Revised

“Line 210: "within" should be removed before "by"”

- Revised

“Line 228: "a" should be changed to "the"”

- Revised

“Line 233: "the" should be added before "size"”

- Revised

“Line 261: "were not" should be changed to "weren't"”

- This appears to be at odds with the lack of contractions used in previous PLOS One publications.

“Line 228: spring-spawned”

- Revised

“Line 229: fall-spawned”

- Revised

“The results aren’t discussed continuously.”

- Revised. Different parts of the results were segmented.

“Line 293: fall-spawned”

- Revised

“Line 293: spring-spawned”

- Revised

“Line 296: there shouldn't be a "the" before fall”

- Revised

“Line 305: fall-spawned”

- Revised

“Line 310: fall-spawned”

- Revised

“Line 314: fall-spawned”

- Revised

“Line 316: "a" larger sample”

- Revised

“Line 330: fall-spawned”

- Revised

“Line 345: "to" should be changed to "of"”

- “representation” is one of the three R’s being referred to. The sentence reads “Given the importance of X, Y, and Z to the overall conservation status…”

“The discussion is not focused on describing the problem, however, there is a weak and disorganized discussion of the topic.”

- Revised. Rearranged and revised the content of the discussion to clarify the objectives and conclusions of the study. We did not set out to necessarily identify and solve a problem, but rather to test for the purported existence of something. Once the test was complete, we described the implications of the results, including new management problems that may exist.

“Does ploidy have an impact on assessment of rigour of your group assignments? Do you view Gulf sturgeon as 2N or 4N? What is the impact of a putative chromosome count of 120 on your interpretation of your FST values? I think a mention of this would help. Does a higher ploidy/chromosome count strengthen your argument?”

- Revised. Added information to the methods explaining that these loci are functionally diploid (Lines 134-135).

“Line 168: you referred to a study from 2022 that 4 adult sturgeon were captured in spawning condition. Were they running milt and/or expressing eggs? Clarification of this would be helpful.”

- Revised (Lines 167-168)

---

## [Editor Report · Decision Letter 1]

9 May 2025

Genetic evidence for a fall spawning group of Gulf sturgeon (*Acipenser oxyrinchus desotoi* ) in the Apalachicola River, Florida, USA

PONE-D-25-04780R1

Dear Dr. Zona,

We’re pleased to inform you that your manuscript has been judged scientifically suitable for publication and will be formally accepted for publication once it meets all outstanding technical requirements.

Kind regards,

Sayyed Mohammad Hadi Alavi

Academic Editor

PLOS ONE

Additional Editor Comments (optional):

Thank you very much for your revision. Publication office may contact you to include copyright for the map used in Figure 1, and to convert Table S1 and Table S2 to word format.
---

## [Editor Report · Acceptance letter]

PONE-D-25-04780R1

PLOS ONE

Dear Dr. Zona,

I'm pleased to inform you that your manuscript has been deemed suitable for publication in PLOS ONE. Congratulations! Your manuscript is now being handed over to our production team.

Kind regards,

on behalf of

Dr. Sayyed Mohammad Hadi Alavi

Academic Editor

PLOS ONE